# Consultation rate and chlamydia positivity among ethnic minority clients at STI clinics in the Netherlands

S. B. Ostendorf[1], C. J. G. Kampman[2]*, C. J. P. A. Hoebe[3,4], J. van der Velden[5], J. L. A. Hautvast[5], C. H. M. van Jaarsveld[5]

1 Public Health Service Gelderland Midden, Arnhem, The Netherlands, 2 Public Health Service Twente, Enschede, The Netherlands, 3 Public Health Service South Limburg, Heerlen, The Netherlands, 4 Department of Social Medicine and Medical Microbiology, Maastricht University Medical Center (MUMC+), Faculty of Health, Medicine and Life Sciences, Care and Public Health Research Institute Maastricht, Maastricht, The Netherlands, 5 Department of Primary and Community Care, Radboud University Medical Centre, Radboud Institute for Health Sciences, Nijmegen, The Netherlands

* k.kampman@ggdtwente.nl

**Data Availability Statement:** All relevant data are within the paper and its Supporting Information files.

## Abstract

### Objectives

Although ethnic minority clients (EMs) from STI endemic countries have a higher risk for STI, little is known about their STI clinic consultation rate proportionality. The aim of this study was to assess consultation and chlamydia positivity rates among different EMs visiting STI clinics in the Netherlands.

### Methods

We calculated consultation rates in EM groups by dividing the number of STI consultations by the total number of inhabitants in the region belonging to an EM, then compared the EM rates to native Dutch rates. Factors associated with chlamydia positivity were analysed using multivariate regression analysis.

### Results

A total of 23,841 clients visiting an eastern Netherlands STI clinic between 2011 and 2013 were included in the analysis, of which 7% were EMs. The consultation rate of native Dutch clients was 22.5 per 1000, compared to 8.5 per 1000 among EMs. Consultation rates in all EMs were lower than in Dutch clients, except for Antillean or Aruban EMs and Latin American EMs.

The chlamydia positivity rate among all clients was 15.5%, and Antillean or Aruban ethnicity (27.1%) EMs had the highest rates. Multivariate analysis identified the following factors associated with chlamydia positivity: Eastern or Northern European EM, African EM, Antillean or Aruban EM, STI related symptoms, heterosexual preference, partner in a risk group, receiving a partner notification, and having had three or more partners in the past six months.

**Funding:** The authors received no specific funding for this work.

**Competing interests:** The authors have declared that no competing interests exist.

## Conclusion

On a population level, most EMs visit STI clinics less often than native Dutch clients, but they have a higher rate of positive chlamydia diagnoses. STI clinics should increase outreach activities for EM clients because they are insufficiently reached by current practices, but contribute substantially to chlamydia incidence rates.

## Introduction

Countries or continents endemic for sexually transmitted infections (STI) and HIV include Turkey, North Africa and Morocco, Suriname, Netherlands Antilles and Aruba, Eastern Europe, Sub-Saharan Africa, Latin America and Asia [1]. In Western countries, STIs are usually more prevalent in people originating from these countries or continents (e.g., ethnic minorities, EMs) than in the rest of the population [2]. For example, the United Kingdom EMs have been over-represented in STI statistics for more than a decade [3–5]. One British study found a significant association between ethnic origin and reported STIs in the previous five years, with increased risk in sexually active black Caribbean and African men compared with white men, and black Caribbean women compared with white women [4]. Another British study of the general population found that STI diagnoses were higher in black Caribbean men (8%) and mixed ethnicity women (7%) than white British participants (4% in men and 3% in women) [6]. A Dutch study found that 12% of Dutch clients visiting an STI clinic were STI positive, with EM clients showing higher positivity rates, ranging from a relative risk of 1.14 in Asian minorities to 1.81 in Antillean/Aruban minorities compared to native Dutch [7]. Another Dutch study reported that EM STI positivity was highest among Latin American and Eastern Europe men who have sex with men, and heterosexual men from the Netherlands Antilles/Aruba (all 25% STI positivity, compared to 20% STI positivity among native Dutch who visited an STI clinic) [1].

Some studies assessed EM clients' number of STI clinic consultations to identify the relative proportion of EM groups attending STI clinics for testing. A British study showed increasing numbers of clients from Central and East Europe attending STI clinics following the European Union expansion [8]. Another British study reported that South Asians, especially women, were reluctant to seek STI clinic care [9]. In 2017, 68% of the STI clinic attendees in the Netherlands were of Dutch origin. Most EM STI clinic consultations involved clients from Asia (5%; 7,667/150,593), followed by Surinamese (5%; 7,536/150,593) and Netherlands Antilles/Aruba clients (3%; 4,087/150,593) [1]. Although these studies assessed the number of consultations for different EMs, most did not assess the proportional representation of EM populations at STI clinics. One Dutch study reported proportionally higher consultation rates among some EMs (Eastern Europeans, Sub-Sahara Africans, Surinamese, Netherlands Antilles/Arubans and Latin Americans) and lower rates among others (Turkey, North Africa, Asia, and Western countries). However, this study did not exclude multiple consultations by the same individual [7], resulting in a potential overestimation of EMs' STI clinic consultations. To inform STI clinic policy, better insight regarding representation of EM clients at STI clinics is needed. Therefore, the current study aimed to assess to what extent the STI clinics in the eastern part of the Netherlands reach EM clients by comparing the total number of STI clinic attendees (including first consultations only, per person per year), to the total number of EM inhabitants. Furthermore, chlamydia positivity was compared between EM clients and clients with a Dutch origin, controlling for behavioural and STI risk factors. The study findings can be used

to develop methods for assessing proportionality of EM STI clinic consultation rates compared to a native population group. Furthermore, the findings regarding EM group representation combined with knowledge of chlamydia positivity rates, can be used to help STI clinics develop more effective outreach policies and activities.

## Methods

### Study design and population

We performed a cross-sectional study using convenience samples from five STI clinics operating within the Public Health Services in the eastern part of the Netherlands. All clients age 15 to 25 years who visited one of the STI clinics during the study period (2011 to 2013) were selected for inclusion. Clients who were 25 years or older were excluded, because Dutch clients of that age cannot consult the STI clinic without any additional STI risk factors, whereas EM clients may consult regardless of their age and STI risk factors.

### Consultation rates

STI clinic data and population data from the Dutch Central Bureau of Statistics were used to establish STI clinic consultation rates per 1000 people, calculated by dividing the number of first consultations in a year of STI clinic attendees belonging to a specific EM (cumulative over the three study years) by the total number of inhabitants (age 15–25) belonging to that EM in the region (cumulative over the three study years) multiplied by 1000. We calculated 95% confidence intervals for these rates using standard methods for single rates (http://vassarstats.net/prop1.html).

### Chlamydia positivity

Chlamydia positivity, determined by a laboratory diagnosis of chlamydia following any consultation at the STI clinic during the study period, included chlamydia diagnoses in all consultations, including repeat consultations, within one year.

### Ethnic minorities

The following EMs were identified: Dutch (The Netherlands), Asian (Asia, excluding Turkey), Western (West and South Europe, United States, Canada, Virgin Islands, Newfoundland, Greenland, Belize and Oceania), East and North European (Bulgaria, former Soviet Union, former Yugoslavia, Hungary, Poland, Romania, Slovakia, Czech Republic, Denmark, Finland, Iceland, Ireland, Luxembourg, Norway, Sweden, Switzerland), African (Africa, excluding Egypt and Morocco), Latin American (North and South America, excluding the United States and Canada), Antillean and Aruban (Netherlands Antilles and Aruba), Turkish (Turkey), Egyptian and Moroccan (Egypt and Morocco), Surinamese (Suriname) and ethnicity unknown (country of birth was not a mandatory question in the client registration system) [1].

EM group assignment was based on the person's own birth country combined with the parental birth country. Persons were defined as belonging to an EM when one or both parents were born outside the Netherlands. When clients' parents were from different EMs, the client was categorised under the mother's birth country; when clients and parents were from different EMs, the client was categorised under his or her own birth country.

### Demographic and behavioural STI risk factors

Demographic variables extracted from the STI database included: gender (women, men, transgender), and ethnic group; behavioural and STI risk factors included: sexual preference (heterosexual, homosexual, or bisexual), number of partners in the last six months (fewer than three, or three or more), STI related symptoms, an STI notification involving a partner, partner in a risk group (men who have sex with men or a partner with an ethnic origin). The risk factors 'intravenous drug use' and 'sex work' were not included in the analysis because of their low numbers.

### Data analysis

First, descriptive statistics were compiled for demographic and behavioural variables and STI risk factors. Consultation rates in specific ethnic groups were compared to native Dutch rates using the absence or presence confidence interval overlap. Chlamydia positivity was calculated separately for each ethnic group. Gonorrhoea, syphilis, hepatitis B, and HIV were not included in the analysis because of low numbers.

Furthermore, binary logistic regression analyses were performed to identify determinants associated with chlamydia positivity. Cases with missing data were excluded from the analysis. Determinants were analysed using univariate analysis and multivariate backward logistic regression, in which a $p < 0.05$ was considered statistically significant. Analyses were conducted using IBM SPSS Statistics for Windows, version 21.0 (IBM Inc., Somers, New York, United States).

### Privacy and ethics

The data were obtained from the medical records of five STI clinics in a fully anonymised and de-identified manner. Medical employees and the STI clinic data manager were responsible for anonymising the dataset used in this study, so the researchers had no access to patient identifying information.

The study protocol was exempted from formal medical-ethical approval under prevailing law in the Netherlands (law of medical scientific research in humans: https://wetten. overheid.nl/BWBR0009408/2020-01-01) because of the retrospective observational design using anonymous data from a patient registration system, as stated by the National Central Committee for Human Studies: www.ccmo.nl and the guidelines for conduct of good behaviour in research www.federa.org. Therefore, individual client consent was not required.

## Results

### Population data

The population of 15 to 25 year-olds between 2011 and 2013 remained relatively stable in the eastern part of the Netherlands, with between 387,318 and 387,924 inhabitants in this age group. The proportion of EMs in the population also remained stable, with 16.5% having an ethnicity other than Dutch in 2011, and 16.8% in 2013.

### STI clinic data

In total, the STI clinics were consulted 26,590 times between 2011 and 2013. This included 2,749 repeat consultations within the same calendar year by the same person, which were excluded, leaving 23,841 clients in the analysis. Seven percent of the clients were EMs. Of all clients, 62% were women and 38% men (Table 1).

**Table 1. Demographic characteristics of 23,841 clients visiting an STI clinic in the eastern part of the Netherlands in 2011–2013 and consultation rates per 1000 person years by ethnicity.**

| | | N (%) | Consultation rate[#] (95%CI) |
|---|---|---|---|
| **First consultations** | Total | 23,841 (100) | - |
| **Gender** | Women | 14,829 (62.2) | - |
| | Men | 9,009 (37.8) | |
| | Transgender | 3 (<0.1) | |
| **Ethnicity** | Dutch | 21,753 (91.2) | 22.5 (22.1–22.7) |
| | EM total | 1643 (7.0) | 8.5 (8.1–8.9)^ |
| | *Asian* | 350 (1.5) | 8.0 (7.2–8.9)^ |
| | *Western* | 334 (1.4) | 8.3 (7.6–9.3)^ |
| | *East/North European* | 283 (1.2) | 13.9 (12.4–15.6)^ |
| | *African* | 214 (0.9) | 18.5 (16.2–21.2)^ |
| | *Latin American* | 163 (0.7) | 30.5 (26.3–35.8)^ |
| | *Antillean/Aruban* | 162 (0.7) | 19.8 (16.9–23.1) |
| | *Turkish* | 67 (0.3) | 1.6 (1.2–2.0)^ |
| | *Egyptian /Moroccan* | 36 (0.2) | 2.7 (1.9–3.8)^ |
| | *Surinamese* | 34 (0.1) | 4.0 (2.8–5.7)^ |
| | Unknown | 435 (1.8) | - |

\# Number of STI clinic consultations per 1000 person years.

- Not applicable.

^ Significantly different from Dutch ethnicity.

Abbreviations: 95%CI = 95% confidence interval.

The native Dutch clients' consultation rate was 22.5 per 1000 person years, compared to a mean rate of 8.5 per 1000 among EMs. Consultation rates in clients originating from Netherlands Antilles and Aruban ethnicity (19.8 per 1000 person years) were comparable to those in Dutch clients. Latin American clients consulted the STI clinic more often than Dutch clients (30.5 vs 22.5 per 1000 person years). The STI clinic consultation rate was significantly lower in all other ethnic groups than in Dutch ethnicity clients. Turkish and Egyptians/Moroccan clients had the lowest consultation rates (1.6 and 2.7 per 1000 person years, respectively).

## Chlamydia positivity

The chlamydia positivity rate of all consultations (including the 2,749 repeated consultations) during the study period was 15.5%. The highest chlamydia positivity rate was found among Antillean and Aruban clients (27.1%), followed by African clients (24.7%). Dutch ethnicity clients had a 15.4% chlamydia positivity rate. The lowest chlamydia positivity rates were observed among Latin Americans (12.0%) and Western Europeans (13.8%), see Table 2. Women and men had similar chlamydia positivity rates (15.5% and 15.6%, respectively), while clients with a heterosexual preference had a higher chlamydia positivity rate (16.4%) than clients with a homosexual or bisexual preference (10.3% and 10.6%, respectively). Clients who had received a partner STI notification had the highest chlamydia positivity rate (41.8%), followed by clients who had STI related symptoms (23.0%).

The multivariable analysis showed that clients more likely to have a positive chlamydia diagnosis than native Dutch clients included those with an Eastern or Northern European ethnicity (OR = 1.44, 95%CI 1.03 to 2.02), an African ethnicity (OR = 1.73, 95%CI 1.25 to 2.39) or an Antillean or Aruban ethnicity (OR = 1.95, 95%CI 1.37 to 2.77). Clients who had STI related symptoms (OR = 2.18, 95%CI 2.02 to 2.35), a heterosexual preference (homosexual preference OR = 0.50,

**Table 2. Chlamydia positivity and determinants of chlamydia positivity in 26,590 consultations at STI clinics in the eastern part of the Netherlands, 2011–2013.**

| | | Number of consultations N (%) | Chlamydia positive % (Nc) | Univariate analysis OR (95%CI) | Multivariate analysis OR (95%CI)* |
|---|---|---|---|---|---|
| **Ethnicity** | Dutch | 24,234 (92.8) | 15.4 (3,721) | 1 | 1 |
| | Asian | 377 (1.4) | 15.6 (59) | 1.02 (0.77–1.35) | 0.84 (0.61–1.14) |
| | Western | 390 (1.5) | 13.8 (54) | 0.89 (0.66–1.18) | 0.95 (0.70–1.30) |
| | East/North European | 349 (1.3) | 17.5 (61) | 1.17 (0.88–1.54) | **1.44 (1.03–2.02)** |
| | African | 239 (0.9) | 24.7 (59) | **1.81 (1.34–2.43)** | **1.73 (1.25–2.39)** |
| | Latin American | 191 (0.7) | 12.0 (23) | 0.76 (0.49–1.17) | 0.74 (0.46–1.19) |
| | Antillean/Aruban | 192 (0.7) | 27.1 (52) | **2.05 (1.49–2.82)** | **1.95 (1.37–2.77)** |
| | Turkish | 81 (0.3) | 18.5 (15) | - | - |
| | Egyptian/ Moroccan | 39 (0.1) | 23.1 (9) | - | - |
| | Surinamese | 47 (0.2) | 23.4 (11) | - | - |
| **Gender** | Women | 16,460 (61.9) | 15.5 (2555) | 1 | 1 |
| | Men | 10,127 (38.1) | 15.6 (1582) | 0.99 (0.93–1.06) | 0.94 (0.87–1.02) |
| | Transgender | 3 (0.0) | 0 | - | - |
| **STI symptoms** | No | 18,589 (69.9) | 12.9 (2404) | 1 | 1 |
| | Yes | 7,418 (27.9) | 23.0 (1706) | **2.03 (1.89–2.17)** | **2.18 (2.02–2.35)** |
| | Missing | 583 (2.2) | - | - | - |
| **Sexual preference** | Heterosexual | 23,692 (89.1) | 16.4 (3876) | 1 | 1 |
| | Homosexual | 1,738 (6.5) | 10.3 (179) | **0.57 (0.48–0.67)** | **0.50 (0.41–0.60)** |
| | Bisexual | 686 (2.6) | 10.6 (73) | **0.61 (0.47–0.78)** | **0.58 (0.44–0.75)** |
| | Missing | 474 (1.8) | - | - | - |
| **Partner in risk group** | No | 22,315 (83.9) | 15.3 (3416) | 1 | 1 |
| | Yes | 3,839 (14.4) | 18.7 (716) | **1.27 (1.16–1.39)** | **1.27 (1.15–1.40)** |
| | Missing | 436 (1.6) | - | - | - |
| **Number of partners last 6 months** | <3 | 15,734 (59.2) | 14.9 (2341) | 1 | 1 |
| | > = 3 | 9,785 (36.8) | 17.6 (1719) | **1.22 (1.14–1.31)** | **1.30 (1.21–1.41)** |
| | Missing | 1,071 (4.0) | - | - | - |
| **Partner STI notification** | No | 22,360 (84.1) | 11.4 (2559) | 1 | 1 |
| | Yes | 3,744 (14.1) | 41.8 (1564) | **5.61 (5.19–6.06)** | **5.99 (5.52–6.50)** |
| | Missing | 486 (1.8) | - | - | - |

N Number of consultations

Nc Number of consultations positive for chlamydia

* All determinants were entered in the multivariable analyses, except for Turkish, Egyptian/Moroccan en Surinamese EM groups due to low numbers

In bold: OR that are significant (at p<0.05)

1 reference

- Not applicable

Abbreviations: OR = odds ratio, 95%CI = 95% confidence interval

bisexual preference OR = 0.58), partners in a risk group (OR = 1.27, 95%CI 1.15 to 1.40), received a partner notification (OR = 5.99, 95%CI 5.52 to 6.50) or had three or more partners in the past six months (OR = 1.30, 95%CI 1.21 to 1.41) were more likely to be chlamydia positive.

## Consultation rates combined with chlamydia positivity

Fig 1 combines the findings from Tables 1 and 2 and displays consultation rates and chlamydia positivity categorised by EM. Clients with an Eastern or Northern European, African,

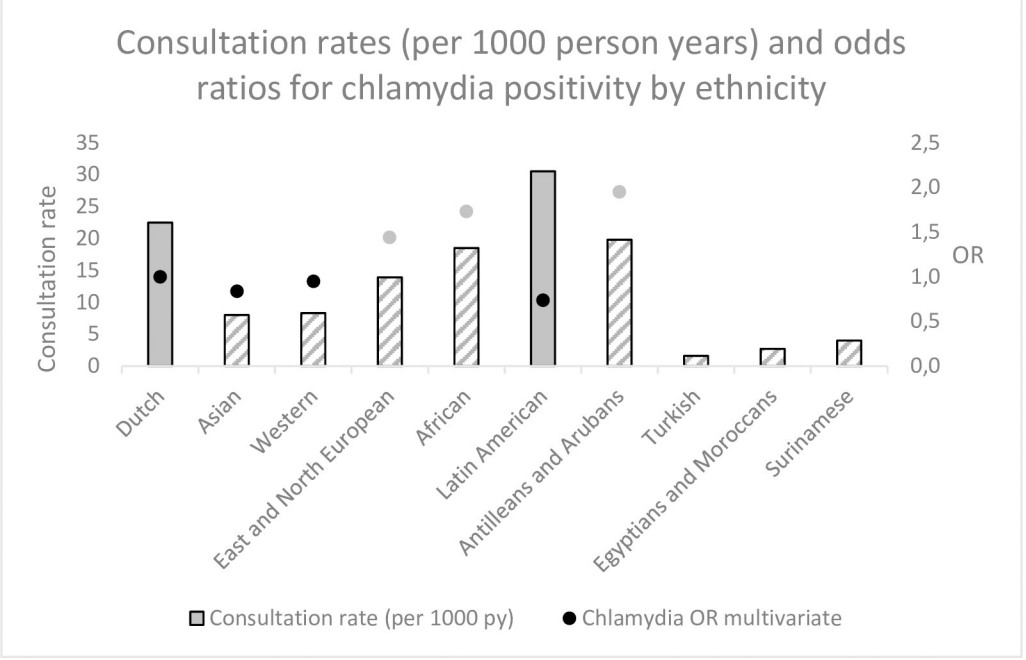

**Fig 1. Consultation rates and chlamydia odds ratios from multivariable analysis by ethnicity.** Consultation rates significantly lower or equal to Dutch rates are striped and chlamydia OR significantly higher than Dutch is coloured grey. For significance see Tables 1 and 2.

Antillean, or Aruban ethnicity had higher chlamydia odds ratios but lower consultation rates than native Dutch clients. Clients with an Asian or Western ethnicity had lower consultation rates and slightly lower chlamydia odds ratios that native Dutch clients (Table 1).

## Discussion

Our study shows that EMs visiting the STI clinic are underrepresented compared to Dutch clients. Western, African, Turkish, Surinamese, East and North European, Asian, and Egyptian and Moroccan clients consult the STI clinic less often than Dutch clients. In contrast, Latin American clients visited more often than Dutch clients and clients from the Netherlands Antilles and Aruba visited the STI clinic equally as often as Dutch clients.

In addition, clients with Netherlands Antillean and Aruban, African, and East and North European ethnicity were chlamydia positive more often than Dutch clients. Considering both consultation rates and chlamydia positivity rates, clients with Antillean and Aruban, African, and East and North European ethnicity are particularly important target groups for regional public health services in the eastern Netherlands, because these EM groups had lower or equal STI clinic consultation rates combined with higher chlamydia positivity rates than native Dutch clients.

### Strengths and limitations

One strength of this study is the high applicability of the proposed innovative approach. Combining population data and STI clinic data to provide proportional person-based consultation rates for different ethnicities is an easy and effective method to inform STI clinic policy.

A second strength is the low risk of bias, since we included a study population of 15- to 25-year-old EM clients whose STI clinic entry requirements matched those applied to the

native Dutch population, and outcome variable measures were uniformly assessed. Our results apply to the eastern part of the Netherlands, and are not generalisable to other Netherlands regions because previous efforts to reach EM groups and provide intervention can differ between STI clinics. However, the method used to calculate EM-specific consultation and STI rates is of value for STI clinics both inside and outside the Netherlands.

One limitation of our study is lack of generalizability to all STIs, since HIV, syphilis, hepatitis B, and gonorrhoea positivity numbers were small, resulting from the age-category limitation.

Another limitation concerns case-ascertainment, because the STI clinics did not collect client residence information, so we therefore assumed that all STI clinic visitors lived locally. Thus, we cannot rule out that some consultations included in the analyses belonged to clients living outside the region.

A third limitation is that we only included data on ethnic minorities receiving STI care from STI clinics, whereas general practitioners also provide STI care in the Netherlands; this limits the generalizability of our results [1]. It is possible that EMs visit general practitioners more often, perhaps because of health system knowledge or travel distance. Conversely, testing at STI clinics is free, but general practitioners charge for testing, which may affect clients' choice of medical facilities.

## Comparison with other studies

Our study's method of calculating consultation rates best compares with the study by van Oeffelen et al. [7], who conducted a similar study in four big Netherlands' cities. Their study differed from ours in that they included repeat consultations in their rates and included a wider age range. In general, van Oeffelen et al. found much higher consultation rates, from two (Dutch) to 12 (Surinamese) times as high as in our study. This might be explained by the two differences described above, plus their STI clinics' policy of focusing more on EMs attending their clinics.

Consultation rates for the ethnicities in our study were similar to those in van Oeffelen et al.'s, except for the Dutch (ranking 1 in our study and 6 in van Oeffelen et al.'s) and the Surinamese (ranking 8 in our study, and 3 in van Oeffelen et al.'s). The higher ranking of the Dutch in our study may be explained by the Dutch and EM's equal access to the STI clinics, stemming from the age restriction we imposed for inclusion. The Surinamese's higher consultation rate in van Oeffelen et al.'s study may be related to different ethnic backgrounds in the Surinamese between the two studies. Oudhof demonstrated that the Suriname population in Amsterdam comprises more African Surinamese, whereas the Surinamese in our study region comprises more South-Asian Surinamese [10]. As Hulstein et al. showed, Surinamese with a South-Asian origin demonstrate lower sexual healthcare seeking behaviour compared to Surinamese with an African origin, which might explain the difference in Surinamese EM consultation rates between the two studies [11].

Our study found that most ethnic minorities have a lower STI clinic consultation rate than the Dutch. However, another Dutch study found that ethnic minorities visit a general practitioner for STI testing as often as or more often than the Dutch, except for the Turkish [12]. In a questionnaire study by Goenee, Surinamese and Antilleans were found to test more often and Turkish and Moroccans less often than the Dutch [13]. This was confirmed by Hulstein et al. [11].

Potential explanations why EM groups have lower consultation rates remain unclear. This study did not include potential barriers that may explain why EM groups have lower consultation rates, such as health literacy, knowledge of health systems, health seeking behaviour,

affordability (e.g. travel cost to the clinic), individual health concerns or language barriers. These factors have been shown to differ between ethnicities or social groups and influence healthcare access [14, 15]. Further research is needed to elucidate which factors set EM groups apart, and shape consultation rates and service access in distinctive ways. This information will be important in tackling the lower consultation rates among specific EM groups.

We found that Netherlands Antilleans and Aruban, African, and East and North European ethnicities were more often chlamydia positive than the Dutch. Van Oeffelen et al. found STI positivity in all ethnicities more often than in the Dutch. Conversely, they did not find higher chlamydia positivity in East Europeans, which was consistent with Visser et al., who also did not find higher chlamydia positivity for African ethnicity [1]. Netherlands Antillean ethnicity had higher chlamydia positive rates in all three studies. Given the low numbers, we could not analyse Surinamese ethnicity in our study. However, van Oeffelen et al., Visser et al., and Hulstein et al. found that clients with Surinamese ethnicity were more often chlamydia positive than the Dutch [1, 7, 11].

High chlamydia positivity in an EM group visiting an STI clinic does not necessarily indicate a high chlamydia prevalence in the regional EM population, because high-risk people may be more likely to visit an STI clinic. Therefore, low chlamydia positivity in an EM may indicate a true low chlamydia prevalence in the regional EM population, but could also indicate an under-representation at the STI clinic of the high-risk group in this EM, or a preference for consulting with a general practitioner.

## Conclusion

In this study we demonstrated a highly applicable and innovative method for calculating consultation rates proportional to local EM populations and provided comprehensive data on STI clinic consultation and chlamydia positivity rates. Our method was shown to be robust and simple, and to provide useful information to inform STI clinic policy and regional interventions. The results indicate that clinics should prioritise reaching out to ethnicities with a low consultation rate and a high positivity rate.

## Supporting information

**S1 Dataset.**
(XLSX)

## Acknowledgments

We would like to acknowledge the participating STI clinics of the Regional Public Health Services in the Eastern part of the Netherlands for providing us access to the anonymised data from their region.

## Author Contributions

**Conceptualization:** S. B. Ostendorf.

**Data curation:** S. B. Ostendorf, J. L. A. Hautvast, C. H. M. van Jaarsveld.

**Formal analysis:** S. B. Ostendorf, C. H. M. van Jaarsveld.

**Methodology:** S. B. Ostendorf, J. L. A. Hautvast, C. H. M. van Jaarsveld.

**Supervision:** J. L. A. Hautvast, C. H. M. van Jaarsveld.

**Writing – original draft:** S. B. Ostendorf, C. J. G. Kampman.

**Writing – review & editing:** C. J. G. Kampman, C. J. P. A. Hoebe, J. van der Velden, J. L. A. Hautvast, C. H. M. van Jaarsveld.

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
