## [Decision Letter · Decision Letter 0]

28 Aug 2020

PONE-D-20-24409

Consultation rate and chlamydia positivity among ethnic minority clients at STI clinics in the Netherlands

PLOS ONE

Dear Dr. Kampman,

Thank you for submitting your manuscript to PLOS ONE. After careful consideration, we feel that it has merit but does not fully meet PLOS ONE’s publication criteria as it currently stands. Therefore, we invite you to submit a revised version of the manuscript that addresses the points raised during the review process. The reviewer recommends minor revisions, mainly of the text.

We look forward to receiving your revised manuscript.

Kind regards,

David M. Ojcius

Academic Editor

PLOS ONE

Journal Requirements:

Reviewers' comments:

Reviewer's Responses to Questions

**Comments to the Author**

1. Is the manuscript technically sound, and do the data support the conclusions?

Reviewer #1: Partly

2. Has the statistical analysis been performed appropriately and rigorously? 

Reviewer #1: Yes

3. Have the authors made all data underlying the findings in their manuscript fully available?

Reviewer #1: Yes

4. Is the manuscript presented in an intelligible fashion and written in standard English?

Reviewer #1: No

5. Review Comments to the Author

Reviewer #1: Congratulations on conducting an interesting study, the findings of which will be interesting to others. I have some comments for you to consider:

1. The English expression needs revision and improvement, particularly in the discussion and all the subsequent sections. Can you have a native English speaking academic review it for you?

2. I have some concerns about your lack of attention to any other factors (con-founders) that may have influenced the rate of attendance at these clinics. For example; EM health literacy levels vs the Dutch community health literacy levels, EM knowledge of the health system vs the dutch community, how long the EM has resided in the Netherlands may influence this and so on.

3. In your strengths you state "A second strength is the low chance of bias as we included a study population of 15 to 25 year olds, who had equal access to the STI clinic for native as well as EM.." I would dispute this statement - there may be enormous bias. I strongly encourage you to explore the concept of 'access' as this will include at least 6 domains where equality may not exist, for example; socio-economic status - how affordable are these clinics to all the various groups? Are they geographically accessible via affordable public transport?, how well do they accommodate the EM? Are they culturally safe for EM? how proficient in the language being used by staff are the EM? Are interpreters freely and easily available? And so on. I think you must, at the very least, inform the reader that these aspects have been considered in the context of your results. The fact that EM attend general practice more readily leads me to ask 'why is this?' and is general practice more "accessible"? I think the discussion needs to be more robust, rather than just summarizing the findings.

Please correct the following typos:

Line 6: remove the word 'over' before more

Line 32: leads - add the 's'

Line 34: change the word 'which' to 'what' extent

Line 38: Change the word in to 'into'

Line 39: Take the 's' off consultations

Line 140: add the words 'had the' before the words consultation rates

Table 2: you must indicate why some values are in bold font (I can guess but shouldn't have to)

6. PLOS authors have the option to publish the peer review history of their article (what does this mean?). If published, this will include your full peer review and any attached files.

Reviewer #1: No

---

## [Author Response · Author response to Decision Letter 0]

28 Jan 2021

Dear Dr David M. Ojcius,

Thank you for reviewing our manuscript entitled “Consultation rate and chlamydia positivity among ethnic minority clients at STI clinics in the Netherlands” for publication in PLOS ONE. We greatly appreciate the time and effort spent on assessing our manuscript. We have revised the manuscript in response to the comments from the reviewer. In this letter, we provide a point-by-point response to these comments and an explanation on the revisions made in the manuscript. 

Comment Reviewer #1: 

Congratulations on conducting an interesting study, the findings of which will be interesting to others. I have some comments for you to consider:

Response: Thank you for reviewing our paper and for giving us a chance to improve our manuscript. We will address each point below. 

Comment 1. The English expression needs revision and improvement, particularly in the discussion and all the subsequent sections. Can you have a native English speaking academic review it for you?

Response: All sections of the paper have been reviewed by a native English speaking academic. We have contacted the editorial office for advice regarding making the changes visible with tracked changes or not. Following this advice the spelling/language improvements are not shown with tracked changes because this would make it difficult to see the changes that were included based on the reviewer’s suggestions.

Comment 2. I have some concerns about your lack of attention to any other factors (con-founders) that may have influenced the rate of attendance at these clinics. For example; EM health literacy levels vs the Dutch community health literacy levels, EM knowledge of the health system vs the dutch community, how long the EM has resided in the Netherlands may influence this and so on.

Response: We recognise the importance of this feedback. The listed factors are indeed relevant, and could be seen as intermediate factors that may explain why EM groups have lower consultation rates. We included the following section in the discussion on page 12 to elaborate on this important issue:

“Potential explanations why EM groups have lower consultation rates remain unclear. This study did not include potential barriers that may explain why EM groups have lower consultation rates, such as health literacy, knowledge of health systems, health seeking behaviour, affordability (e.g. travel cost to the clinic), individual health concerns or language barriers. These factors have been shown to differ between ethnicities or social groups and influence healthcare access [14, 15]. Further research is needed to elucidate which factors set EM groups apart, and shape consultation rates and service access in distinctive ways. This information will be important in tackling the lower consultation rates among specific EM groups.”

Comment 3. In your strengths you state "A second strength is the low chance of bias as we included a study population of 15 to 25 year olds, who had equal access to the STI clinic for native as well as EM.." I would dispute this statement - there may be enormous bias. 

Response: In our response to the previous comment we have included the issue of barriers to healthcare service access. In the section, the reviewer refers to (page 11) we address a different aspect of access. What we meant here (in the strengths and limitations section) was that the same entry criteria to access the STI clinic applied to native as well as EM. In the Netherlands, anyone in the age of 15 to 25 years may consult a STI clinic for free. In contrast, for those over 25 years, access criteria differ: as native Dutch who are over 25 cannot consult the STI clinic without any additional STI risk factors, whereas EM clients who are over 25 have free access regardless of STI risk factors. We therefore restricted our analyses to the 15-25 year olds.

We rephrased the sentence to make it more clear, in the strengths and limitations section on page 11 as follows:

“A second strength is the low risk of bias, since we included a study population of 15- to 25-year-old EM clients whose STI clinic entry requirements matched those applied to the native Dutch population, and outcome variable measures were uniformly assessed.” 

Comment 3. (continued) I strongly encourage you to explore the concept of 'access' as this will include at least 6 domains where equality may not exist, for example; socio-economic status - how affordable are these clinics to all the various groups? Are they geographically accessible via affordable public transport?, how well do they accommodate the EM? Are they culturally safe for EM? how proficient in the language being used by staff are the EM? Are interpreters freely and easily available? And so on. I think you must, at the very least, inform the reader that these aspects have been considered in the context of your results. The fact that EM attend general practice more readily leads me to ask 'why is this?' and is general practice more "accessible"? I think the discussion needs to be more robust, rather than just summarizing the findings.

Response: In response to comment 2 (see above), we have addressed these important issues by adding a paragraph to the discussion on page 11 (please see out response to comment 2).

Comment 4. Please correct the following typos:

Line 6: remove the word 'over' before more

Line 32: leads - add the 's'

Line 34: change the word 'which' to 'what' extent

Line 38: Change the word in to 'into'

Line 39: Take the 's' off consultations

Line 140: add the words 'had the' before the words consultation rates

Response: We (and the native speaker) have corrected the typos.

Comment 4 (continued) Table 2: you must indicate why some values are in bold font (I can guess but shouldn't have to).

Response: We added the following text in the footnote of Table 2 to explain why some values are in bold: “In bold: OR that are significant (at p<0.05).”

As part of this rebuttal we reviewed our paper, and found an inconsistency that we corrected. On page 6 in the Methods section we corrected the statement as follows:

“Determinants were analysed using univariable analysis and multivariable backward logistic regression, in which a p<0.05 was considered statistically significant.”

We greatly appreciate your time and effort for reading our revisions and hope that our responses provided above have satisfactorily addressed the reviewer’s questions and comments. 

Sincerely, on behalf of all co-authors,

C.J.G. Kampman

Public Health Service Twente

P.O. Box 1400

7500 BK Enschede

The Netherlands

k.kampman@ggdtwente.nl

+31 534876768

---

## [Editor Report · Decision Letter 1]

2 Feb 2021

Consultation rate and chlamydia positivity among ethnic minority clients at STI clinics in the Netherlands

PONE-D-20-24409R1

Dear Dr. Kampman,

We’re pleased to inform you that your manuscript has been judged scientifically suitable for publication and will be formally accepted for publication once it meets all outstanding technical requirements.

Kind regards,

David M. Ojcius

Academic Editor

PLOS ONE
---

## [Editor Report · Acceptance letter]

4 Feb 2021

PONE-D-20-24409R1 

Consultation rate and chlamydia positivity among ethnic minority clients at STI clinics in the Netherlands 

Dear Dr. Kampman:

I'm pleased to inform you that your manuscript has been deemed suitable for publication in PLOS ONE. Congratulations! Your manuscript is now with our production department. 

Kind regards, 

on behalf of

Dr. David M. Ojcius 

Academic Editor

PLOS ONE